# Molecular Foundations of Inflammatory Diseases: Insights into Inflammation and Inflammasomes

## Mi Eun Kim and Jun Sik Lee *

Department of Biological Science, Immunology Research Lab & BK21-Four Educational Research Group for Age-Associated Disorder Control Technology, Chosun University, Gwangju 61452, Republic of Korea; kimme0303@chosun.ac.kr
* Correspondence: junsiklee@chosun.ac.kr; Tel.: +82-062-230-6651

**Abstract:** Inflammatory diseases are a global health problem affecting millions of people with a wide range of conditions. These diseases, including inflammatory bowel disease (IBD), rheumatoid arthritis (RA), osteoarthritis (OA), gout, and diabetes, impose a significant burden on patients and healthcare systems. A complicated interaction between genetic variables, environmental stimuli, and dysregulated immune responses shows the complex biological foundation of various diseases. This review focuses on the molecular mechanisms underlying inflammatory diseases, including the function of inflammasomes and inflammation. We investigate the impact of environmental and genetic factors on the progression of inflammatory diseases, explore the connection between inflammation and inflammasome activation, and examine the incidence of various inflammatory diseases in relation to inflammasomes.

**Keywords:** inflammation; inflammatory diseases; inflammasome; signaling pathway

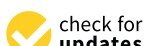



## 1. Introduction

Inflammatory diseases are characterized by a complex interplay of genetic factors, environmental triggers, and dysregulated immune responses [1–3]. In terms of the importance of innate immunity, innate immunity serves as the first line of defense against invading pathogens and foreign bodies [4]. It depends on a wide range of immune cells that have pattern recognition receptors (PRRs), such as neutrophils, dendritic cells, and macrophages. PRRs recognize pathogen-associated molecular patterns (PAMPs) and damage-associated molecular patterns (DAMPs) [5–7]. The identification of these biological signals triggers off a variety of immunological reactions that ultimately result in inflammation [8]. Understanding the complex signaling pathways and molecular components within the innate immune system is the key to understanding the causes of inflammatory diseases. Several factors contribute to inflammation [9–11]. The intricate genetic factors contributing to the risk of diseases like Crohn's disease within Inflammatory Bowel Disease (IBD) have been revealed through genome-wide association studies (GWASs). These studies have identified specific mutations in genes associated with inflammatory diseases [12]. In addition, environmental triggers, particularly disruptions in the gut microbiome, significantly contribute to disease development. The significance of microbial interactions in shaping disease pathophysiology is demonstrated by the enduring connection between dysbiosis, characterized by an imbalance in microbial diversity, and various inflammatory diseases [13,14]. A major factor in the progression of inflammatory diseases from acute to chronic inflammation is the dysregulated immune response mediated by PRRs such as toll-like receptors (TLRs) and nucleotide oligomerization domain (NOD)-like receptors [15]. The convergence of genetic and environmental factors with immune dysregulation underscores the complexity of inflammatory diseases. A thorough understanding is needed to enable targeted treatment approaches for inflammatory diseases. This review also examines the role of inflammasomes in the pathogenesis of various inflammatory diseases. In recent years, the regulation

of inflammatory responses by inflammasomes has been a focus of interest. Inflammasomes are multi-protein complexes responsible for the activation of pro-inflammatory cytokines, including interleukin (IL)-1β and IL-18. These molecular factors are essential for controlling immune responses and include AIM2 (absent in melanoma 2) and NOD-like receptors (NLRs) [16–18]. The activation of inflammasomes in response to pathogens, endogenous signals, or toxins indicates the importance of inflammasomes in determining the course of inflammatory diseases [19]. To elucidate the pathogenesis of these diseases, it is essential to understand the complex molecular composition and activation mechanisms of inflammasomes. These inflammasomes, present in immune cells such as macrophages, control important processes such as pyroptosis, cytokine release, and caspase-1 activation. Inflammasomes react to different danger signals in specific diseases including gout, rheumatoid arthritis, and diabetes, leading to tissue damage and inflammation [19–21]. Therefore, this review focuses on the complex molecular mechanisms of inflammasome activation and inflammation. This review describes the association of inflammasomes with inflammatory responses in inflammatory diseases and the importance of their regulation in the control of inflammatory diseases. The main keywords of the review are various inflammatory diseases caused by inflammatory responses and their association with inflammasomes to regulate them. Additionally, the review analyzes the regulatory mechanisms and examines appropriate works, referencing the most recently reported articles. Furthermore, the review elaborates on their involvement in chronic inflammation, fibrosis, and tissue repair across various diseases. Understanding these complex pathways is essential for developing targeted therapeutic strategies to effectively treat inflammatory diseases.

## 2. Inflammatory Diseases Factors

### 2.1. Genetic Factors

Genetic factors significantly influence an individual's capacity to develop inflammatory diseases [22]. Recent developments in genetic research, particularly through GWAS, have identified various susceptibility genes and single nucleotide polymorphisms (SNPs) associated with inflammatory diseases [23,24]. These discoveries help us to understand the complex genetic background to inflammatory disease. For instance, specific genetic alterations have been identified as contributors to the development of IBD. Gene variants such as *ATG16L1* and *NOD2* have been associated with an increased risk of Crohn's disease (CD) [25,26]. A total of 163 genetic loci have been linked to IBD, which includes ulcerative colitis (UC) and CD, according to previous GWASs. Sixty of these loci exhibited varying effects on both CD and UC, while 50 showed no difference between the two groups [27]. The remaining 53 loci were specific to either UC or CD. Remarkably, 113 of these loci were also found to be associated with other complex features, such as mycobacterial infections and immune-mediated diseases. Recent research has identified 38 novel genetic loci associated with IBD risk. These studies used immunochip genotyping data from a variety of cohorts, including East Asian, European, and Indian [27–31]. Of these unique loci, the majority are involved in the development of both CD and UC, whereas the remaining 11 loci are specific to either CD or UC. The first gene related to IBD was NOD2, and certain mutations in its coding area, such as L1007fs, R702W, and R703C, were connected to CD in patients [22]. These *NOD2* mutations cause a loss of protein function, raising the possibility that bacterial infections and the development of IBD are related to compromised innate immune responses. Conversely, rare diseases including Blau Syndrome and early-onset sarcoidosis are linked to gain-of-function mutations in the NOD2 NATCH domain. R334W and R334Q are the most frequent mutations, accounting for 40–80% of patient cases [32,33]. Several GWASs have also been performed in pediatric IBD patients, showing correlations with genes associated with autophagy, innate immunity, and inflammatory cytokine production. Furthermore, correlations with G protein subunit alpha 12 and hepatocytes nuclear factor 4 alpha suggest impaired epithelial barrier function. Additionally, studies have associated both types of IBD to hypomorphic X-box binding protein 1 modifications, where deletions in intestinal epithelial cells result in spontaneous enteritis [22]. Notably, the genetic

profiles of IBD patients differ between countries, with correlations with specific genes (*ATG16L2*, *DUSP5*, and *TBC1D1*) rather than ATG16L1 and NOD2 [28,34]. Whole Exome Sequencing (WES), one of the most recent developments in genetic analysis technology, has increased the capacity to find new genetic connections with CD. Missense mutations in the T and B cell-expressed *PRDM1* gene and frequent changes in the *NDP52* gene, an adaptor protein involved in the selective autophagy of intracellular bacteria, have been found via WES investigations [35]. In addition, WES has identified unique mutations in the *DLG1* gene in some patients from this country that have been linked to T-cell polarity, T-cell proliferation, and T-cell receptor signaling, all of which are risk factors for CD [36,37]. These genetic markers demonstrate the complexity of the genetic basis of inflammatory diseases, which often involve multiple susceptibility genes with minimal effects on each isolated gene (Table 1). It is becoming increasingly clear that there are other factors beyond genetic predisposition that affect a person's health. Genetic research has not only identified susceptibility genes, but has also provided insights into the control of immune responses. Genetic differences can affect how different immune cells work, how pro-inflammatory cytokines are produced, and how the immune system responds to infection. Investigating the complex relationship between immunological dysregulation and genetic factors is the objective of the developing control of immunogenetics. Understanding these genetic factors is essential for creating personalized treatment plans that consider each patient's own genetic profile as well as for advancing our understanding of the relationship between innate immunity and inflammatory diseases.

**Table 1.** Summary of inflammatory diseases caused by genetic factors.

| Factor | Target | Pathway | Ref. |
|---|---|---|---|
| Genetic Factors | Susceptibility Genes, SNPs | Inflammatory Disease Development | [23,24] |
| NOD2 | Increased Risk of CD | Increased Risk of CD | [25,26] |
| 38 Novel Genetic Loci | IBD Risk | Development of CD and UC | [27–31] |
| *NOD2* Mutations (e.g., L1007fs, R702W, R703C) | CD in Patients | Compromised Innate Immune Responses | [22] |
| GWAS in Pediatric IBD Patients | Autophagy, Innate Immunity, Inflammatory Cytokine Production | Impaired Epithelial Barrier Function | [22] |
| X-box Binding Protein 1 Modifications | Both Types of IBD | Spontaneous Enteritis | [22] |
| Whole Exome Sequencing (WES) | T and B cell-expressed *PRDM1* gene, *NDP52* gene | CD, T-cell Polarity, T-cell Proliferation, T-cell Receptor Signaling | [35] |
| Unique Mutations in *DLG1* Gene | T-cell Polarity, T-cell Proliferation, T-cell Receptor Signaling | Risk Factors for CD | [36,37] |

### 2.2. Environmental Triggers

Environmental factors contribute significantly to the development and aggravation of inflammatory diseases [38,39]. The gut microbiota has generated a lot of interest among these factors. Billions of bacteria contribute to the gut microbiota, which is essential for maintaining intestinal homeostasis and has a major effect on immune system regulation [40,41]. Dysbiosis has been consistently associated with several inflammatory diseases, including IBD [42]. In order to understand how microbial metabolites, changes in mucosal barrier function, and interactions with immune cells affect the pathophysiology of disease, researchers are actively exploring the complicated interactions between the gut microbiota and the host immune system [41,43,44].

In the context of the gastrointestinal system, PRRs such as TLRs, NOD, and NLRs play a critical role in host defense and in the regulation of inflammation [15,45]. NLR proteins represent a diverse group of cytoplasmic PRRs with multiple functions, including

their involvement in the formation of inflammasomes responsible for the maturation of specific interleukins [20]. The development of IBD has been linked to the dysregulation of certain NLRs. Recent research has shown significant changes in the mRNA levels of several different NLRs in the colons of IBD patients [45–47]. Furthermore, previous studies have shown that several types of NLR family members (NLRP1, NLRP3, NLRP6, and NLRC4) have the potential to significantly alter the immune response for IBD [45,48]. The NLRP3 inflammasome of the innate immune system is an essential component that triggers caspase-1 activation and the release of pro-inflammatory cytokines such as IL-1β and IL-18 in response to microbial infection and progressive cell damage. NLRP3 inflammasome activation has been shown to be influenced by the composition of the gut microbiome in IBD [49]. Polymorphisms in the NLRP3 gene have also been linked to a higher risk of CD in humans [50]. However, there is an ongoing discussion regarding the specific processes involved in the contribution of NLRP3 to epithelial barrier integrity and mucosal inflammation. For instance, as demonstrated in animal studies, NLRP3 expression in macrophages has been linked to both a reduction in inflammation and an increase in the pathology of experimentally induced colitis [45,51]. The activation of NLRP3 in hematopoietic cells, such as macrophages, may provide protection against disease in IBD. On the other hand, the mediation of disease susceptibility also depends on the earliest onset of the NLRP3 inflammasome in non-hematopoietic cells, especially intestinal epithelial cells. In addition, the synthesis of inflammatory mediators such as IL-6, CXCL-1, and tumor necrosis factor (TNF)-α has been associated with NLR activation in intestinal epithelial cells [52,53].

NLRP6, a member of the NLR family, has been proposed to function as part of an inflammasome due to its ability to interact with the adaptor protein ASC (apoptosis-associated speck-like protein containing a CARD). NLRP6 is a recently implicated NLR in the pathobiology of IBD [54]. Ectopic expression studies have demonstrated the ability of human NLRP6 to cooperate with ASCs and promote inflammasome formation [55]. Both resident macrophages and epithelial cells in the gastrointestinal tract exhibit significant levels of NLRP6 expression. In the colon, NLRP6 demonstrates robust protective effects consistent with its expression pattern and the function of other NLRs that generate inflammasomes. Reduced levels of IL-18 in the colon result in a significant decrease in epithelial cell repair, and animals deficient in NLRP6 experience both spontaneous and dextran sodium sulfate (DSS)-induced experimental colitis, which is exacerbated. Furthermore, NLRP6-deficient mice show increased tumorigenesis in the azoxymethane/dextran sodium sulfate (DSS) model, associated with reduced epithelial cell repair and increased proliferation [45,54,56,57].

Interestingly, investigations involving bone marrow chimeras have revealed that while colitis severity appears to be associated with NLRP6 function in the epithelial cell compartment, higher carcinogenesis in NLRP6-deficient mice has been associated with the hematopoietic cell compartment. Compared to wild-type animals, mice deficient in NLRP6 have a considerably different commensal microbiome composition; therefore, the gut microbiota directly affects the pathophysiology of a disease [58]. The chemokine CCL5, which stimulates inflammation and epithelial cell proliferation through the IL-6 pathway, is expressed by the microbiota and is linked to both colitis and colitis-associated carcinogenesis [59–61]. Studies involving fecal transplantation and co-housing have demonstrated that NLRP6-deficient mice can transmit both cancer and colitis susceptibility to wild-type animals [62]. A wild-type microbiome can also protect NLRP6-deficient animals from inflammatory bowel disease. Salmonella typhimurium, Escherichia coli, and Listeria monocytogenes are all extremely resistant in Nlrp6-deficient animals, according to additional characterization studies [63–65]. Nonetheless, the bacterial phylum Bacteroidetes (Prevotellaceae), which is the principal representative of the microbiota-associated phenotype and has a significant connection to the pathophysiology of IBD, has significantly expanded in these animals [45]. Recent research has shown that impaired goblet cell autophagy in NLRP6-deficient mice leads to inaccurate exocytosis of goblet cell mucin granules [66–68].

These results suggest that Prevotellaceae are able to colonize atypical sites within the crypts of NLRP6-deficient animals due to this impairment in mucin synthesis, resulting in persistent infections leading to colitis. However, despite this important study demonstrating the critical role of NLRP6 in regulating microbial ecology in mice, there are currently no studies translating these findings to human populations, and further research is needed to determine the underlying mechanism by which NLRP6 regulates the microbiota (Table 2).

**Table 2.** Summary of the impact of environmental triggers on inflammation and inflammasome regulation.

| Factor | Target | Pathway | Ref. |
|---|---|---|---|
| Gut Microbiota | Dysbiosis | Immune System Regulation, Inflammatory Disease Pathophysiology | [40,41] |
| PRRs (e.g., TLRs, NOD, NLRs) | NLR Dysregulation | Host Defense, Inflammation Regulation | [45] |
| NLRP3 | Gut Microbiome Composition, Polymorphisms | Inflammasome Activation, IL-1β and IL-18 Release | [49] |
| NLRP6 | ASC Interaction, IL-18 Expression | Inflammasome Formation, Epithelial Cell Repair | [54] |
| CCL5 | Inflammation, Epithelial Cell Proliferation | IL-6 Pathway | [59–61] |
| Lifestyle, Pollutant Exposure, Infectious Pathogens | Immune Response, Inflammation | Disease Onset and Progression | [63–65] |

In addition to the microbiome, a variety of environmental factors influence inflammation and immune regulation. The immune response and the progression of inflammation are influenced by a variety of factors, including lifestyle, exposure to pollutants, and exposure to infectious pathogens. The specific methods by which these environmental factors affect the beginning of disease are currently being studied.

*2.3. Dysregulated Immune Responses*

Inflammation is the immune system's response to harmful stimuli, such as pathogens, toxins, and damaged cells. The dysregulation of the immune response is a central feature of various inflammatory diseases [2]. Its main responsibility is to remove these risks and initiate the healing process. On the other hand, acute inflammation can become chronic and contribute to the onset of a variety of chronic inflammatory diseases as long as it is not properly controlled. Swelling, pain, heat, and loss of tissue function are typical symptoms of inflammation at the tissue level. These appearances result from immune, vascular, and inflammatory cell responses to infections or injuries.

Changes in vascular permeability, the release of inflammatory mediators, and in the recruitment and accumulation of leukocytes are important microcirculatory processes that occur throughout the inflammatory process. Several pathogenic triggers, both common and non-infectious, can induce inflammation [2,69]. These factors trigger a cascade response of chemical signals that initiates repair responses in the tissues that have been damaged. As a result, leukocytes from the general circulation are attracted to the areas of inflammation, where they generate cytokines and trigger responses of inflammation. The general procedure involves the coordinated activation of signaling pathways that modulate the levels of inflammatory mediators between native tissue cells and inflammatory cells collected from the circulation of blood. PRRs, such as TLRs, can recognize endogenous signals associated with tissue or cell damage, known as DAMPs, as well as microbial structures, known as PAMPs. Host biomolecules identified as DAMPs have the ability to initiate and sustain non-infectious inflammatory responses. In the absence of infection, damaged cells can also attract innate inflammatory cells by producing DAMPs. Inflammatory responses are mediated by various PRR families, including TLRs, retinoic acid-inducible gene (RIG)-I-like

receptors, C-type lectin receptors, and NLRs. The thoroughly researched TLR family of PRRs plays an essential role in triggering the inflammatory response [7,70,71]. There are currently approximately 13 identified members of the TLR family. Activating intracellular signaling cascades through TLRs results in the nuclear translocation of transcription factors, including nuclear factor kappa-B (NF-κB), interferon regulatory factor 3, and activator protein-1 [72]. Interestingly, DAMPs and PAMPs share receptors, including TLR4, suggesting that infectious and non-infectious inflammatory responses are similar. Common inflammatory mediators and regulatory mechanisms are involved in the development of many chronic diseases through their influence on inflammatory pathways. Microbial products and cytokines such as IL-1β, IL-6, and TNF-α provide primary inflammatory stimuli by interacting with specific receptors, triggering inflammation. Important intracellular signaling pathways, such as the NF-κB, Janus kinase (JAK)-signal transducer and activator of transcription (STAT), mitogen-activated protein kinase (MAPK) pathways, and extracellular signal-regulated kinases (ERK), are activated by these receptor activations [2,52].

In clinical applications, inflammatory indicators have been used to evaluate the effects of therapeutic interventions and to differentiate between beneficial and harmful biological processes. In addition, they have the potential to be used as predictive indicators of inflammatory processes, linking the causes and outcomes of a variety of diseases, including infection, endothelial dysfunction, and cardiovascular disease. Predictive stimuli stimulate the production of inflammatory cytokines, including IL-1β, IL-6, and TNF-α, as well as inflammatory proteins and enzymes by inflammatory cells, especially macrophages and adipocytes [2,73,74]. Cytokines, predominantly released from immune cells like monocytes, macrophages, and lymphocytes, play a crucial role in regulating inflammation. They may facilitate or inhibit inflammation, respectively, potentially acting as pro-inflammatory or anti-inflammatory factors. Cytokines responsible for inflammation include chemokines, interleukins, transforming growth factors (TGFs), interferons (IFNs), and TNFs, among others [75]. They are mostly produced by cells to attract leukocytes to the site of damage or infection. Although cytokines assist in controlling the immune system's response to infection and inflammation, excessive levels of cytokines might result in tissue damage, hemodynamic abnormalities, organ damage, and, in the final stages, death. The treatment of inflammatory diseases and the identification of stimuli-mediated inflammation depend on an understanding of the regulation of cytokine pathways.

In the event of oxidative stress or infection, inflammatory proteins and enzymes present in the blood, such as alpha-1-acid glycoprotein, haptoglobin, serum amyloid A, fibrinogen, and C-reactive protein, help to restore homeostasis and inhibit microbial development without the aid of antibodies [2,76]. Enzymes such as inducible nitric oxide synthase (iNOS), cyclooxygenase (COX)-2, glutathione peroxidase (GPx), NADPH oxidase, high-mobility group box 1 (HMGB1), and superoxide dismutase (SOD) are among those whose abnormal activation has been linked to the development of inflammation-related diseases, including cancer and cardiovascular disease [77]. For example, extracellular HMGB1 can trigger TLR-coupled signaling pathways, mainly targeting TLR4, leading to the production of pro-inflammatory cytokines such as IL-1β and TNF-α [78].

In medical applications, inflammatory proteins and enzymes have been used as biomarkers for infection and inflammation. Oxidative stress is influenced by antioxidant defense mechanisms, which are additional indications of inflammation. Reactive oxygen species (ROS), malondialdehyde (MDA), and 8-hydroxy-2-deoxyguanosine can all be generated in response to increased oxidative stress. These substances can all activate different transcription factors, such as STAT, AP-1, and NF-κB phosphorylation [79]. Genes encoding chemokines, inflammatory cytokines, and growth factors can all have their levels of expression increased by this process. Since oxidative stress is associated with the development of several diseases, including atherosclerosis, diabetes, cancer, cardiovascular disease, and hypertension, oxidative stress products are suitable indicators of the inflammatory response [2,79,80].

Monocytes, activated macrophages, and other cell types that mediate specific responses to tissue damage and infection belong to a variety of cell types that are involved in a carefully coordinated network during inflammatory responses. Along with chemokines and growth factors that recruit neutrophils and monocytes, injured epithelium and endothelial cells release factors at the site of tissue injury to initiate the process of inflammatory responses [2,81]. Monocytes are attracted into tissue damage by chemotaxis and have the ability to develop into macrophages and dendritic cells. Chronic inflammatory diseases, diabetes, asthma, atherosclerosis, cancer, and autoimmune diseases belong to a variety of diseases related to alterations in immune cell activities carried on by inflammation [2,82–84]. As essential components of the mononuclear phagocyte system, macrophages are involved in the initiation, repair, and recovery of inflammation. In addition to phagocytosing and delivering antigens, macrophages additionally affect the immune system during inflammation by producing growth factors and cytokines (Table 3). If the immune response is not efficiently regulated to eliminate harmful triggers, it can lead to various inflammatory diseases.

**Table 3.** Summary of the relationship between immune dysregulation and inflammatory diseases.

| Factor | Target | Pathway | Ref. |
|---|---|---|---|
| Toll-like Receptors (TLRs) | Nuclear factor kappa-B (NF-κB), Interferon regulatory factor 3 (IRF3), Activator protein-1 (AP-1) | TLR activation leads to intracellular signaling cascades | [72] |
| Inflammatory Cytokines (e.g., IL-1β, IL-6, TNF-α) | Various receptors, including TLR4 | Activation of NF-κB, Janus kinase (JAK)-STAT, MAPK pathways | [52] |
| Reactive Oxygen Species (ROS) | Transcription factors such as STAT, AP-1, NF-κB phosphorylation | Induces expression of chemokines, cytokines, and growth factors | [79] |
| Oxidative Stress Products (e.g., ROS, MDA, 8-hydroxy-2-deoxyguanosine) | Transcription factors such as STAT, AP-1, NF-κB phosphorylation | Induce expression of chemokines, inflammatory cytokines, and growth factors | [79,80] |
| Growth Factors and Chemokines | Neutrophils and monocytes | Recruit immune cells to the site of tissue injury | [82–84] |

## 3. Inflammasome and Inflammatory Diseases

### 3.1. Activation in Inflammatory Diseases

Multiprotein complexes within cells called inflammasomes are primarily found in immune cells and include dendritic and macrophages, and these cells have been identified as important players in the pathophysiology of a variety of inflammatory diseases [49]. They are essential for regulating the activation of caspase-1, a proteolytic enzyme. In turn, caspase-1 processes and matures two pro-inflammatory cytokines, IL-1β and IL-18. Additionally, caspase-1 activation enables pyroptosis, an instance of organized cell death [85]. These complexes are also responsible for processing and secreting pro-inflammatory cytokines, including IL-1β and IL-18, in response to a variety of danger signals, such PAMPs and DAMPs [20]. The pathophysiology of a variety of inflammatory diseases has a strong connection with the activation of inflammatory proteins, for example, the activation of NLRP3 inflammasome in the case of gout, a severe arthritic disease based on by the accumulation of uric acid crystals in joints. NLRP3 inflammasome activation has been brought on by the formation of monosodium urate crystals in joints [86,87]. This activation promotes the maturation and release of IL-1β, which in turn induces inflammation, the recruitment of immune cells, and damage to joints. This example shows how inflammasomes mediate the inflammation resulting from endogenous danger signals and the pathophysiology of disease. Procaspase-1, which includes a CARD domain, and ASC, which acts as a bridge connecting NLRP3 to procaspase-1, represent the three essential components of the NLRP3 inflammasome, a necessary component of the body's self-defensive response to infections and cellular stress. Danger signals, which are typically classified

as either PAMPs or DAMPs, drive the trans-Golgi network's NLRP3 inflammasome activation [20,49]. Two separate steps are involved in the activation process. TLRs identify PAMPs and DAMPs to initiate the NF-κB signaling pathway in the first step, which is called the sensing and production step. Precursor proteins such NLRP3, IL-1β, and IL-18 are produced in greater levels as a result of this. The NLRP3 protein, procaspase-1, and ASC combine to produce a mature complex in the second stage, which is referred to as the assembly and effector Stage.

The inflammatory response that results is subsequently carried on by this complex, which transforms proinflammatory cytokines IL-1β and IL-18 into their active forms [20,71,88]. The NLRP3 inflammasome promotes the immune response against many infections through TLR-dependent mechanisms. Activated by microbial stimuli such as lipopolysaccharides, bacterial RNA, and bacterial muramyl dipeptide, the NLRP3 inflammasome plays a critical role in enhancing the immune response [89,90]. Furthermore, non-infectious danger signals such reactive oxygen species (ROS), changes in calcium ion levels, nitric oxide (NO) generation, and mitochondrial dysfunctions are all capable of activating the NLRP3 inflammasome. As a second messenger, mitochondrial ROS may be generated in response to PAMPs or DAMPs [20,91].

For the activation of the NLRP3 inflammasome in the case of muscle atrophy, angiotensin II can increase mitochondrial dysfunction and the production of ROS within the mitochondria, which in turn stimulates the NLRP3 inflammasome. The mitochondria have to operate properly for the NLRP3 inflammasome to be activated. Many factors can lead to mitochondrial dysfunctions, such as alterations in calcium ion levels and NO, which stimulate the release of oxidized mitochondrial DNA in TLR interactions [92–94]. Mitophagy, a process that removes damaged mitochondria, can also impact excessive inflammasome activation. Furthermore, diseases such as asbestosis and silicosis can be caused by the NLRP3 inflammasome. This inflammasome is activated by components like asbestos and silica, which are endocytosed by pulmonary macrophages. Additionally, the accumulation of hydroxyapatite crystals in osteoarthritis and monosodium urate in gout may activate the NLRP3 inflammasome, leading to inflammation and joint disorders [95,96]. The NLRP3 inflammasome induces the release of IL-1β, accelerating the progression of atherosclerosis. Mesenchymal stem cells generated from bone marrow can inhibit the activation of the NLRP3 inflammasome by reducing the activation of caspase-1 and IL-1β. This is achieved through reducing the production of ROS in the mitochondria. Systemic inflammation is exacerbated by the overproduction of IL-1β and IL-18, which are essential for the adaptive immune response [89,97,98].

While the exact mechanisms by which the NLRP3 inflammasome identifies DAMPs remain unclear, it is believed that elements including ROS generation within the mitochondria, $K^+$ efflux, and $Ca^{2+}$ signaling play a part in this process. A variety of factors, including ROS, reperfusion, obesity, and ischemia, have been found to induce the activation of the NLRP3 inflammasome [20,99,100]. Inflammasomes have been related to the etiology of inflammatory diseases, including joint disorders like gout and neurological diseases including Alzheimer's disease. One common aspect of neurodegenerative disorders is chronic neuroinflammation, which is indicated by the presence of pro-inflammatory cytokines and activated immune cells in the brain. The generation of IL-1β, a factor in neuronal damage and cognitive decline, has been associated with the activation of inflammatory enzymes in astrocytes and microglial cells. This demonstrates the complex function of inflammasomes in neuroinflammatory processes and suggests that they may have a role in the development of neurodegenerative disorders [101,102]. Moreover, inflammasomes have become important players in metabolic diseases including type 2 diabetes. Insulin resistance and the dysregulation of glucose homeostasis have been associated with aberrant inflammasome activation, specifically involving the NLRP3 inflammasome. When metabolic stress and high glucose levels activate the NLRP3 inflammasome, IL-1β is secreted, which exacerbates systemic inflammation and reduces insulin sensitivity. This shows the complex relation-

ships present between metabolic alteration, inflammasome activation, and the etiology of metabolic diseases [103,104].

Gout is a condition characterized by inflammation brought on by crystals of monosodium urate (MSU). Despite not normally being present in normal joints, neutrophils play a key role in this inflammation. After MSU crystals are deposited in the joint, the main thing that seems to happen is that they interact with the local joint cells, particularly the synovial lining cells, which then causes neutrophils to infiltrate the region. Mononuclear phagocytes are important in the first response to MSU crystal formation, according to recent in vitro research. Monocyte cell lines that come into contact with MSU crystals release proinflammatory cytokines, including IL-1β. MSU crystal phagocytosis is an essential step in this process. It has also been observed that, as microbial infections, MSU crystals may operate as a "danger" signal to cells. Kim et al. provided evidence of the significance of the NLRP3 inflammasome in detecting MSU accumulation and stimulating the innate immune response [90]. The results of the in vitro study showed that mouse macrophages deficient in several plasma membrane components, including NLRP3, were unable to induce IL-1β in response to MSU. Specifically, these mice showed reduced neutrophil infiltration following intraperitoneal treatment with MSU, indicating that MSU was unable to induce IL-1β production by macrophages from mice lacking certain plasma membrane components [105]. Interestingly, after intraperitoneal MSU infusion, these mice also showed decreased neutrophil infiltration, indicating that the NLRP3 plasma membrane plays an important mediating role between the recognized gout-related stimuli and the additional clinical signs of acute gout attacks [87]. The discovery that MSU and free fatty acids (FFAs) efficiently combine to produce IL-1β revealed the molecular mechanisms underlying the relationship between gout and the excessive consumption of food. The treatment of peripheral blood mononuclear cells from healthy persons was found to enhance the release of IL-1β; however, the presence of FFAs was associated with notable amounts of active IL-1β. It is interesting to note that in response to MSU and FFA infusion, animals lacking caspase-1 and ASC significantly reduced their release of IL-1β. However, mice lacking the *NLRP3* gene did not exhibit the same reduction [106,107].

Considerable evidence suggests that inflammasomes are involved in inflammatory joint diseases including gout, osteoarthritis (OA), and rheumatoid arthritis (RA). Inflammasomes may in fact contribute to OA, according to recent studies. OA joints contain a number of factors that can activate the inflammasome, including hydroxyapatite, MSU, calcium pyrophosphate, uric acid, and calcium phosphate crystals. Pro-inflammatory cytokines including TNF-α and IL-1β can be released when these occurrences activate the NLRP3 inflammasome. The production of these inflammatory cytokines requires inflammasome priming, and, in OA synovial membrane cells, NLRP3 expression was shown to be significantly higher than in normal controls. The majority of cells in the synovial membrane, synovial fibroblasts, produce IL-1β and TNF-α, which induce the production of enzymes that degrade cartilage. These enzymes further degrade type II collagen and proteoglycans, which are important components of cartilage. Examples of these enzymes are matrix metalloproteinases and thrombospondin motif metalloproteinases [86,108]. The process causes collagen and proteoglycan particles to be released into the joint space, which in turn promotes the production of additional pro-inflammatory cytokines, such IL-18, which promote inflammation and stimulate the joint. Angiogenesis, sensory innervation, new bone formation, cartilage fissuring, and additional cascades of events in OA may all be involved in the pain which OA patients have. Moreover, pyroptosis, a kind of cell death, may be involved in the pathogenic alterations observed in OA. Osteophyte production, synovial inflammation, and cartilage degradation are all symptoms of OA, and pyroptosis can cause these symptoms. In addition to the data related to OA, there is a connection between inflammasomes and metabolic diseases including metabolic syndrome and obesity. Adipose tissue macrophages in obesity activate or prime the inflammasome and produce pro-inflammatory cytokines that may accelerate the onset of OA. Increased matrix metalloproteinase (MMP) activity is frequently linked to obesity. MMP activity

products function as DAMPs, further stimulating inflammasomes and inducing cartilage degradation and chondrocyte pyroptosis. Because of this connection, the word "metabolic syndrome-associated OA" was created. Furthermore, uric acid accumulation in obese people may exacerbate long-term low-grade inflammation, which would further relate inflammasomes to the onset of OA. According to recent research, pyroptosis and consequent inflammasome activation may play a major role in the degenerative alterations in the joint. As a result, there is increasing proof to indicate the presence of inflammasomes in the development and progression of OA, even though their direct role in the disease has not been as well investigated as it has in other inflammatory joint diseases such as gout and RA [86,109,110]. A clear correlation exists between the pathophysiology of OA, cytokine production, and inflammasome activation, as supported by evidence from various study levels. In conclusion, to modulate the activation of inflammasomes, it is crucial to understand which specific inflammasomes are involved in each disease and the signaling pathways they activate. The diversity of inflammatory enzyme complexes and their function in modulating immune responses to different risk signals renders them significant in the context of inflammatory diseases.

### 3.2. Dual Role of Inflammasomes in Tissue Damage and Repair across Diseases

Regarding the role of the NLRP3 inflammasome in tissue damage and repair, inflammasomes are essential for the initiation of cytokine production and inflammation; nevertheless, they also have an effect on the processes of tissue damage and repair in various disease events. Inflammasome activation has the potential to aggravate inflammation and tissue damage, though it can also be involved in the process of coordinating tissue repair [17,111]. By promoting the activation of fibroblasts and the excessive deposition of extracellular matrix proteins, inflammasome activation in fibrotic diseases is capable of resulting in constant inflammation and tissue damage. Damage and tissue fibrosis are the outcomes of this procedure. Fibrotic diseases affecting the liver and lungs, among other organs, have been related to the NLRP3 inflammasome in particular. Organ dysfunctions and significant tissue remodeling may result from inflammasome-driven inflammation in certain situations [112,113]. It has been demonstrated that the NLRP3 and other inflammasomes contribute significantly to fibrosis by continuously releasing inflammatory cytokines, including IL-1β and IL-18, which in consequence induce chronic inflammation. TGF-β1 and IL-6, both cytokines that may trigger macrophage polarization, are capable of having an influence on fibroblasts, which are sentinel cells in tissues and play a critical role in regulating macrophage activation. In the presence of fibrosis, the tissue microenvironment may be altered by reciprocal signaling between fibroblasts and macrophages, further promoting fibrosis. In the absence of a wound healing process, activated fibroblasts, known as myofibroblasts, generate an excessive amount of collagen, contributing to fibrosis. Emphysema and pulmonary fibrosis are two disorders where studies have shown the involvement of NLRP3 inflammasomes in fibrosis [112,114].

Adenosine triphosphate, ROS, and uric acid are just a few of the substances that might activate these inflammasomes. The NLRP3 inflammasome has been associated with a number of chronic diseases, including diabetes, fibrosis, gout, Alzheimer's disease, and autoinflammatory diseases. It is also important in pathogen identification. Caspases-1 are activated during the complex interactions that occur during the assembly of the NLRP3 inflammasome with proteins such as ASC and mammalian NIMA-related kinases (NEK)7. The active caspase-1 enzyme facilitates chronic profibrotic diseases by cleaving IL-1β and IL-18 into their active forms. In general, fibrosis is largely caused by the NLRP3 inflammasome, and developing effective treatment plans for fibrotic diseases requires an understanding of how this inflammasome is activated and regulated [112,115,116].

On the other hand, inflammasome activation promotes tissue repair under specific conditions. For example, inflammasome activation integrates inflammatory responses with tissue regeneration in liver regeneration after damage. Liver function is restored by the secretion of IL-1β and IL-18, which are mediated by inflammasomes that promote

hepatocyte proliferation and tissue regeneration. It turns out that inflammasomes have a dual function in tissue damage and repair, which shows how complicated their functions are in various diseases [117]. Homeostasis, inflammation, proliferation, and remodeling are the four phases that make up the foundation of the acute wound healing process. Excessive or prolonged inflammation is a characteristic of chronic wounds that inhibits healing and leaves scars. Innate immunity in the skin is influenced by the NLRP3 inflammasome, which is present in epithelial tissues including the skin. The NLRP3 inflammasome's contribution to multiple phases of cutaneous wound healing is being investigated in recent studies.

The NLRP3 inflammasome and skin wound healing are two processes that mulberry leaf and fruit extract (MLFE) has been demonstrated to regulate. In obese conditions, MLFE appears to normalize NLRP3 levels and decrease skin inflammation during the initial phases of wound healing [118,119]. Nanomaterials with distinct spatial configurations known as tetrahedral framework nucleic acids (TFNAs) have a number of advantageous characteristics, such as anti-inflammatory, antioxidant, anti-fibrotic, angiogenic, and skin-wound-healing activities. It has been demonstrated that TFNAs accelerate wound healing and reduce scarring, indicating their ability to promote skin tissue regeneration. Additionally, by maintaining endothelial cell function via antioxidant activity, TFNAs have the potential to promote diabetic wound healing through processes including angiogenesis and epithelialization. It has been demonstrated that TFNAs decrease skin collagen content, inflammatory cytokine levels, and pyroptosis [120,121].

The interaction between pyroptosis and the inflammasome pathways is still poorly understood, despite the complexity of the wound healing process. To address chronic inflammation and pyroptosis in wound healing and to develop specific treatment strategies, additional research is needed to fully understand the interactions between various signaling pathways during the course of wound healing [119]. Consequently, this implies that controlling the inflammatory response and the inflammasome plays an essential role in treating a variety of inflammatory diseases. In summary, the important function that inflammasomes play in inflammatory diseases is emphasized. In the context of many disease conditions, it shows their function in identifying danger signals, initiating the production of cytokines, and mediating immunological responses. Additionally, depending on the situation, inflammasomes may promote tissue regeneration and repair or aggravate tissue damage through persistent inflammation. For the purpose of creating specific treatments and understanding the complicated pathophysiology of inflammatory diseases, it is essential to understand the specific inflammasomes implicated in each disease as well as their following implications.

## 4. Conclusions

In this review, we discuss the mechanisms of inflammation and the inflammatory diseases they cause. Inflammatory diseases are triggered by various pro-inflammatory factors involving different signaling pathways. We have shown that inflammasomes are critical factors in several diseases, and their modulation may be a key factor in controlling inflammatory diseases. Many studies have investigated inflammatory diseases and strategies/therapies using inflammasomes in atherosclerosis, tumors and the tumor microenvironment, cardiovascular dysfunction, and have highlighted their importance. Furthermore, recent studies have shown that inflammasomes have a dual effect, being involved in both tissue damage and repair. Therefore, the regulation of inflammasomes is thought to be an important factor in the control of inflammatory diseases. In conclusion, it is necessary to understand the regulation and signaling of various inflammatory factors, including inflammasomes, in order to understand and control various inflammatory diseases, suggesting the need for continued research and discussion on clinical trials and inflammasomes.

**Author Contributions:** M.E.K. and J.S.L. contributed to and designed the content of the review. M.E.K. wrote the first draft of the manuscript and J.S.L. edited the text. All authors have read and agreed to the published version of the manuscript.

**Funding:** This research was funded by the National Research Foundation of Korea (NRF), funded by the Korean government (MIST, No. NRF-2020R1A2C1012984).

**Institutional Review Board Statement:** Not applicable.

**Informed Consent Statement:** Not applicable.

**Data Availability Statement:** The data presented in this study are available on request from the corresponding author. The data are not publicly available due to privacy.

**Conflicts of Interest:** The authors declare no conflicts of interest.

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
