# Peer review of "Molecular Foundations of Inflammatory Diseases: Insights into Inflammation and Inflammasomes"

_cimb, doi:10.3390/cimb46010030_

Round 1

Reviewer 1 Report

Comments and Suggestions for Authors

The rewiew is devoted to interesting and actual theme - molecular mechanisms of in flammatory diseases. The review quotes mainly recent publications so it is actual for specialists. Nexertheless a giant amount of informations leads to difficulties of understanding this review. I propose to illustrate some parts of review by adding scheme:

for activation of NLRP3 inflammosome in case of gout, severe arthritic (p.7, lines 302-326),

for activation of NLRP3 inflammosome in case of muscle atrophy (p.7, lines from 327),

for role of NLRP3 inflammosome in tissue damage (p.9) and repair (p.10).

I'm convinced, this will make easier to understand these processes.

Also it will be usefull to add description of conflicting data in this area found in the literature. 

Author Response

Response for Reviewer I

The rewiew is devoted to interesting and actual theme - molecular mechanisms of inflammatory diseases. The review quotes mainly recent publications so it is actual for specialists. Nexertheless a giant amount of informations leads to difficulties of understanding this review. I propose to illustrate some parts of review by adding scheme:

Comment 1: for activation of NLRP3 inflammosome in case of gout, severe arthritic (p.7, lines 302-326),

for activation of NLRP3 inflammosome in case of muscle atrophy (p.7, lines from 327),

for role of NLRP3 inflammosome in tissue damage (p.9) and repair (p.10).

I'm convinced, this will make easier to understand these processes.

Also it will be usefull to add description of conflicting data in this area found in the literature.

Response: Thank you for your helpful comment. We have revised the manuscript as requested by the reviewer.

Reviewer 2 Report

Comments and Suggestions for Authors

The paper provides a comprehensive review of the molecular mechanisms underlying inflammatory diseases, with a particular focus on the role of inflammasomes. It discusses how genetic and environmental factors, along with dysregulated immune responses, contribute to the development of these diseases. Below are the comments:

1.       The paper discusses the role of genetic and environmental factors in inflammatory diseases. However, it's unclear whether the paper includes the most recent research findings in these areas. The review should be updated with more recent research findings to ensure the information is current and complete.

2.       While the paper mentions understanding the regulation and signaling of various inflammatory factors has been implicated as a critical factor in controlling inflammatory diseases in conclusion, it might benefit from a more extensive discussion on the current state of these strategies/treatments, their effectiveness, implications of the findings and more future research directions.

3.       While it was mentioned in the abstract that complicated interaction between genetic variables, environmental stimuli, and dysregulated immune responses shows the complex biological foundation of various diseases, it might benefit from a more detailed discussion on the mechanism of these interactions.

4.       Some paragraphs(sections 2.2, 2.3, 3.1, 3.2) are too long, it might be beneficial to divide them into several paragraphs to make them easy to follow and read.

Author Response

Response for Reviewer II

The paper provides a comprehensive review of the molecular mechanisms underlying inflammatory diseases, with a particular focus on the role of inflammasomes. It discusses how genetic and environmental factors, along with dysregulated immune responses, contribute to the development of these diseases. Below are the comments:

Comment 1: The paper discusses the role of genetic and environmental factors in inflammatory diseases. However, it's unclear whether the paper includes the most recent research findings in these areas. The review should be updated with more recent research findings to ensure the information is current and complete.

Response: Thank you for your helpful comment. We have updated the reference with the latest literature as requested by the reviewer.  

Comment 2: While the paper mentions understanding the regulation and signaling of various inflammatory factors has been implicated as a critical factor in controlling inflammatory diseases in conclusion, it might benefit from a more extensive discussion on the current state of these strategies/treatments, their effectiveness, implications of the findings and more future research directions.

Response: Thank you for your helpful comment. As requested by the reviewer, we have included a discussion of research directions in the conclusion section. Furthermore, we will write a new review paper to discuss the strategies/treatments, effectiveness of the results, and future research directions.

Comment 3: While it was mentioned in the abstract that complicated interaction between genetic variables, environmental stimuli, and dysregulated immune responses shows the complex biological foundation of various diseases, it might benefit from a more detailed discussion on the mechanism of these interactions.

Response: Thank you for your helpful comment. We have incorporated a description of the intricate mechanisms underlying various diseases and included the latest references throughout the manuscript.

Comment 4: Some paragraphs(sections 2.2, 2.3, 3.1, 3.2) are too long, it might be beneficial to divide them into several paragraphs to make them easy to follow and read.

Response: Thank you for your helpful comment. We have split it into several paragraphs to make it easier to read, as requested by the reviewer

Reviewer 3 Report

Comments and Suggestions for Authors

The manuscript “Molecular basis of inflammatory diseases: insights on inflammation and inflammasomes” by Mi Eun Kim and Jun Sik Lee focuses on the molecular mechanisms underlying inflammatory diseases, highlighting the role of inflammasomes and inflammation. It also examines the impact of environmental and genetic factors on the progression of these diseases.

The subject of the work itself is interesting and necessary, but before it is published it still requires significant corrections. Below I present the advantages and disadvantages of this work.

Aspects requiring improvement:

1. The manuscript reviews the existing literature, and I would ask the authors to reflect and search the literature, and perhaps provide more experimental data or original research.

2. Another aspect related to the first point is methodology. I would like to ask the authors on what basis they made this review. What were the search parameters, keywords, and range of years, i.e. criteria for inclusion in the review of specific works. Such information would help readers search for articles on similar topics.

3. Although the article delves deeper into inflammasomes, it may be beneficial to take a broader look at other inflammatory pathways and their interconnections, providing a more holistic view of the field.

4. Another aspect that would enrich the work is to discuss the different inflammasome pathways in the context of different diseases or patient populations, including age, gender, and ethnic differences.

5. The manuscript contains interesting information, but reading the text may discourage you from reading it for a long time. I ask the authors to consider adding figures that would enrich the work (diagrams, diagrams, infographics of mechanisms and interactions).

6. I would also suggest that the authors add a section discussing gaps in current knowledge and proposing specific future research directions.

Advantages of the manuscript:

1. The article discusses the intricate interplay of genetic factors, environmental factors, and immune reactions in inflammatory diseases.

2. The manuscript incorporates the latest research, highlighting the latest discoveries in the field.

3. The article highlights the importance of understanding these mechanisms for the development of targeted therapeutic strategies, which makes them very important for clinical trials.

In summary, the manuscript in its current form is informative and detailed in some respects but requires significant refinement to increase its relevance, depth, and appeal to a wider audience. Deepening the discussion on clinical implications and future research directions, as well as adding visual aspects to the manuscript, could significantly improve its reception.

Author Response

Response for Reviewer II

The manuscript “Molecular basis of inflammatory diseases: insights on inflammation and inflammasomes” by Mi Eun Kim and Jun Sik Lee focuses on the molecular mechanisms underlying inflammatory diseases, highlighting the role of inflammasomes and inflammation. It also examines the impact of environmental and genetic factors on the progression of these diseases. The subject of the work itself is interesting and necessary, but before it is published it still requires significant corrections. Below I present the advantages and disadvantages of this work.

Aspects requiring improvement

Comment 1: The manuscript reviews the existing literature, and I would ask the authors to reflect and search the literature, and perhaps provide more experimental data or original research.

Response: Thank you for your positive review of our manuscript. This review presents evidence supporting the significance of inflammation and the regulation of the inflammasome in various diseases.

Comment 2: Another aspect related to the first point is methodology. I would like to ask the authors on what basis they made this review. What were the search parameters, keywords, and range of years, i.e. criteria for inclusion in the review of specific works. Such information would help readers search for articles on similar topics.

Response: Thank you for your helpful comment. Our review was based on keyword searches for inflammatory responses that cause various diseases and the role of inflammasomes in the regulation of these diseases, including recent articles, and references to published studies.

Comment 3: Although the article delves deeper into inflammasomes, it may be beneficial to take a broader look at other inflammatory pathways and their interconnections, providing a more holistic view of the field.

Response: Thank you for your positive review of our manuscript.

Comment 4: Another aspect that would enrich the work is to discuss the different inflammasome pathways in the context of different diseases or patient populations, including age, gender, and ethnic differences.

Response: Thank you for your positive review of our manuscript. We will discuss the inflammasome pathway in various diseases or patient populations, such as age, gender, and ethnic differences, in our next review article after further research is performed, as there is currently limited information in the literature.

Comment 5: The manuscript contains interesting information, but reading the text may discourage you from reading it for a long time. I ask the authors to consider adding figures that would enrich the work (diagrams, diagrams, infographics of mechanisms and interactions).

Response: Thank you for your helpful comment. To assist the reviewers in reading the manuscript, we have prepared Tables 1, 2 and 3 as requested by the reviewers.

Comment 6: I would also suggest that the authors add a section discussing gaps in current knowledge and proposing specific future research directions.

Response: Thank you for your helpful comment. We presented the future research direction in the in the Conclusions section.

Advantages of the manuscript:

Comment 7: 1. The article discusses the intricate interplay of genetic factors, environmental factors, and immune reactions in inflammatory diseases. 2. The manuscript incorporates the latest research, highlighting the latest discoveries in the field. 3. The article highlights the importance of understanding these mechanisms for the development of targeted therapeutic strategies, which makes them very important for clinical trials.

In summary, the manuscript in its current form is informative and detailed in some respects but requires significant refinement to increase its relevance, depth, and appeal to a wider audience. Deepening the discussion on clinical implications and future research directions, as well as adding visual aspects to the manuscript, could significantly improve its reception.

Response: Thank you for your helpful comment. We have updated the latest research literature and visualized the contents of the papers. Furthermore, future research directions are mentioned in the conclusion section.

Round 2

Reviewer 2 Report

Comments and Suggestions for Authors

The authors have addressed all my comments. 

Author Response

Response for Reviewer II

The authors have addressed all my comments.

Response: Thank you for your helpful comment.

Reviewer 3 Report

Comments and Suggestions for Authors

Dear Authors,

Thank you for taking into account your suggestions and responding to my comments. Work looks much better with tables.

At the same time, I believe the article deserves to be published after adding a materials and methods section, in which the authors would describe in detail what terms they used to search for literature.

In my opinion, this is an essential aspect because it allows the reader to verify the article's content and facilitates further exploration of the topic.

Therefore, as a minor note, I suggest inserting a materials and methods section with specific keywords used to prepare the manuscript.

This is my only comment, but I will leave it for the Editor and you to consider.

I appreciate your cooperation.

Author Response

Response for Reviewer III

Dear Authors,

Thank you for taking into account your suggestions and responding to my comments. Work looks much better with tables.

At the same time, I believe the article deserves to be published after adding a materials and methods section, in which the authors would describe in detail what terms they used to search for literature.

In my opinion, this is an essential aspect because it allows the reader to verify the article's content and facilitates further exploration of the topic.

Therefore, as a minor note, I suggest inserting a materials and methods section with specific keywords used to prepare the manuscript.

This is my only comment, but I will leave it for the Editor and you to consider.

I appreciate your cooperation.

Response: Thank you for your helpful comment. As requested by the reviewer, we have included an explanation in the Data analysis and keyword descriptions section.
